# Methods for assessing seasonal and annual trends in wasting in Indian surveys (NFHS-3, 4, RSOC & CNNS)

Robert Johnston[1], Gaurav Dhamija[2]*, Mudit Kapoor[3], Praween K. Agrawal[4], Arjan de Wagt[1]

1 UNICEF, New Delhi, India, 2 Indian Institute of Technology Hyderabad, Telangana, India, 3 Indian Statistical Institute, New Delhi, India, 4 IPE Global Limited, Delhi, India

* gaurav.dhamija@la.iith.ac.in

**Data Availability Statement:** Two rounds of the National Family Health Survey (NFHS) are a publicly available secondary dataset containing no personally identifiable information. These data can

## Abstract

Wasting in children under-five is a form of acute malnutrition, a predictor of under-five child mortality and of increased risk of future episodes of stunting and/or wasting. In India, national estimates of wasting are high compared to international standards with one in five children found to be wasted. National surveys are complex logistical operations and most often not planned or implemented in a manner to control for seasonality. Collection of survey data across differing months across states introduces seasonal bias. Cross-sectional surveys are not designed to collect data on seasonality, thus special methods are needed to analyse the effect of data collection by month. We developed regression models to estimate the mean weight for height (WHZ), prevalence of wasting for every month of the year for an average year and an overall weighted survey estimates controlling for the socio-demographic variation of data collection across states and populations over time. National level analyses show the mean WHZ starts at its highest in January, falls to the lowest in June/August and returns towards peak at year end. The prevalence of wasting is lowest in January and doubles by June/August. After accounting for seasonal patterns in data collection across surveys, the trends are significantly different and indicate a stagnant period followed by a decline in wasting. To avoid biased estimates, direct comparisons of acute malnutrition across surveys should not be made unless seasonality bias is appropriately addressed in planning, implementation or analysis. Eliminating the seasonal variation in wasting would reduce the prevalence by half and provide guidance towards further reduction in acute malnutrition.

## Introduction

Wasting (WHZ <-2SD) is a temporal condition that provokes serious threats to the health, development and life of a child [1] and is a response to nutrient intakes not meeting demands for physiological and biochemical functions, growth and capacity to respond to illness. In period of deprivation, the body uses body fat, muscle and other nutrients to maintain essential

be accessed from the DHS website on request (https://www.dhsprogram.com/). CNNS and RSOC survey data have not been released in the public domain. Hence we have uploaded both the datasets as Supporting information files. All relevant data of CNNS and RSOC are within the paper and its Supporting information files.

**Funding:** UNICEF has received funding from Lakshmi and Aditya Mittal. However, there are no grant numbers are available to identify funding from them. However, the UNICEF's Internal grant numbers are SC130707 & SC180233. Funding individuals (Lakshmi and Aditya Mittal) have not played a role in the study design, data collection and analysis, decision to publish, or preparation of the manuscript and only provided financial support in the form of authors' salaries and/or research materials.

**Competing interests:** The authors have declared that no competing interests exist.

metabolic processes [2, 3], causing failure to grow or weight loss. Wasting can occur at any stage of development, including in utero [3].

Globally an estimated 13% of under-five child deaths are attributed to wasting each year [4]. In India, 20% of under-five deaths are provoked by child wasting [5]. The mortality risk rises exponentially with the severity of malnutrition [6]. India has achieved consistent improvements in under-five mortality [7]. For those children who survive, each wasting experience increases the risk of stunting [8], which is associated with reduced cognition, educability and economic productivity [9] and causally related to increased risk of high body fat/obesity, cardiovascular disease and increased risk of having low birth weight (LBW) babies in women [10].

Demographic surveys using cross-sectional methods are representative of specific geographic areas and the specific time period of data collection. As seasonality is seldom accounted for during data collection or later analysis, bias is introduced into the reported results. For more rare conditions such as severe acute malnutrition that are difficult to capture in cross-sectional data, this becomes problematic. On a given day, data collection using cross-sectional methods will identify only a small proportion of children with wasting and even fewer of bilateral oedema as compared to those who had experienced the condition in the month(s) prior. To control for seasonal variation, SMART methods recommended seasonally planned surveys with one month recommended maximum length of data collection [11].

Other child growth research has documented the difficulty to capture wasting due to variation over time both within-individuals and within-populations [12, 13]. As discussed above the number of wasting episodes in a population over a year is not captured in cross-sectional, survey-based estimates of wasting. Data from Northern Nigeria comparing survey and coverage data show that the number of children who suffer a period of being wasted within a year (incidence) is several times higher than the number of children who are wasted at one point in time (prevalence) [14]. Other researchers have attempted to identify more appropriate methods for data collection on malnutrition [15].

These issues in data collection and presentation of wasting estimates from cross-sectional surveys have led to questions about the global calculation of the annual burden of the child population affected by wasting. Some researchers considered the number calculated from unadjusted survey estimates (without incidence control factor) a minimum, rather than actual estimate of the annual wasting burden [16]. When comparing new wasting cases that occured over one year compared to cross-sectional estimates in Niger, the number of wasted children tripled [17].

Acute malnutrition has seasonal peaks due to the well-recognized effect of seasonality on food security, caring practices and disease patterns [1]. Data from longitudinal studies show that wasting was highly seasonal with the lowest weight-for-length Z scores during the rainy season, particularly in south Asia. Intra-household analyses of nutritional status in India find that women and children under five are the most vulnerable populations to malnutrition [18] and thus the effects of seasonal wasting. The seasonal effects of malnutrition on women are demonstrated through birth weights but no longer by body mass index or mid-upper arm circumference [19, 20]. In Uttar Pradesh, a longitudinal sample of birth weights showed that infants conceived from April to September were significantly lighter than those conceived from October to March [19].

It has been recognized that seasonality affects wasting prevalence in India and across South Asia [21–23]. India experiences significant variations in agro-climatic conditions throughout the year [24]. Seasonality has a strong effect on temperature, diets [25], food availability, vitamin A deficiency [26], disease [27], labor conditions/incomes [28], water availability and sanitation, caring practices, infant and young child feeding and even health seeking behaviors [29]. Analysis and presentation of results without accounting for seasonal patterns can lead to

misinterpretation [30]. Seasonality encompasses a complex interplay of factors that can heighten the risk of acute malnutrition in children, demonstrated by wasting.

Wasting in India is considered to be a significant public health issue. The direct estimates of wasting decreased from 19.8% with 2005–06 National Family Health Survey-3 (NFHS-3) to 17.3% with 2016–18 Comprehensive National Nutrition Survey (CNNS). Despite significant economic growth and huge investment in nutrition over the past two decades, the trend of wasting since 2005 has not shown a regular declining trend.

Unadjusted analysis of the effects of seasonal patterns on wasting in the 2005–06 NFHS-3 and 2013–14 Rapid Survey on Children (RSOC), 2015–16 National Family Health Survey-4 (NFHS-4) and 2016–18 CNNS surveys presents the strong effect of month of data collection on the estimated prevalence of wasting by the different surveys (S1 Fig).

Unadjusted monthly estimates should not be compared due to both the planned and ad hoc nature of survey data collection. Data collection often starts in urban areas or densely populated rural areas to allow for intense supervision. Coordinators often plan data collection in barren deserts in the cool season and in flood zones outside of the rainy season. During data collection, one month may be spent collecting data from upper class neighborhoods, while the next may be in the poorest slums. In analysis of unadjusted data, months with small sample sizes are likely to provide less representational data compared to the months with larger sample size.

The timing of data collection by month varied greatly across the four surveys (Fig 1). The RSOC data was collected from November 2013 to April 2014 with 86% of data from December,

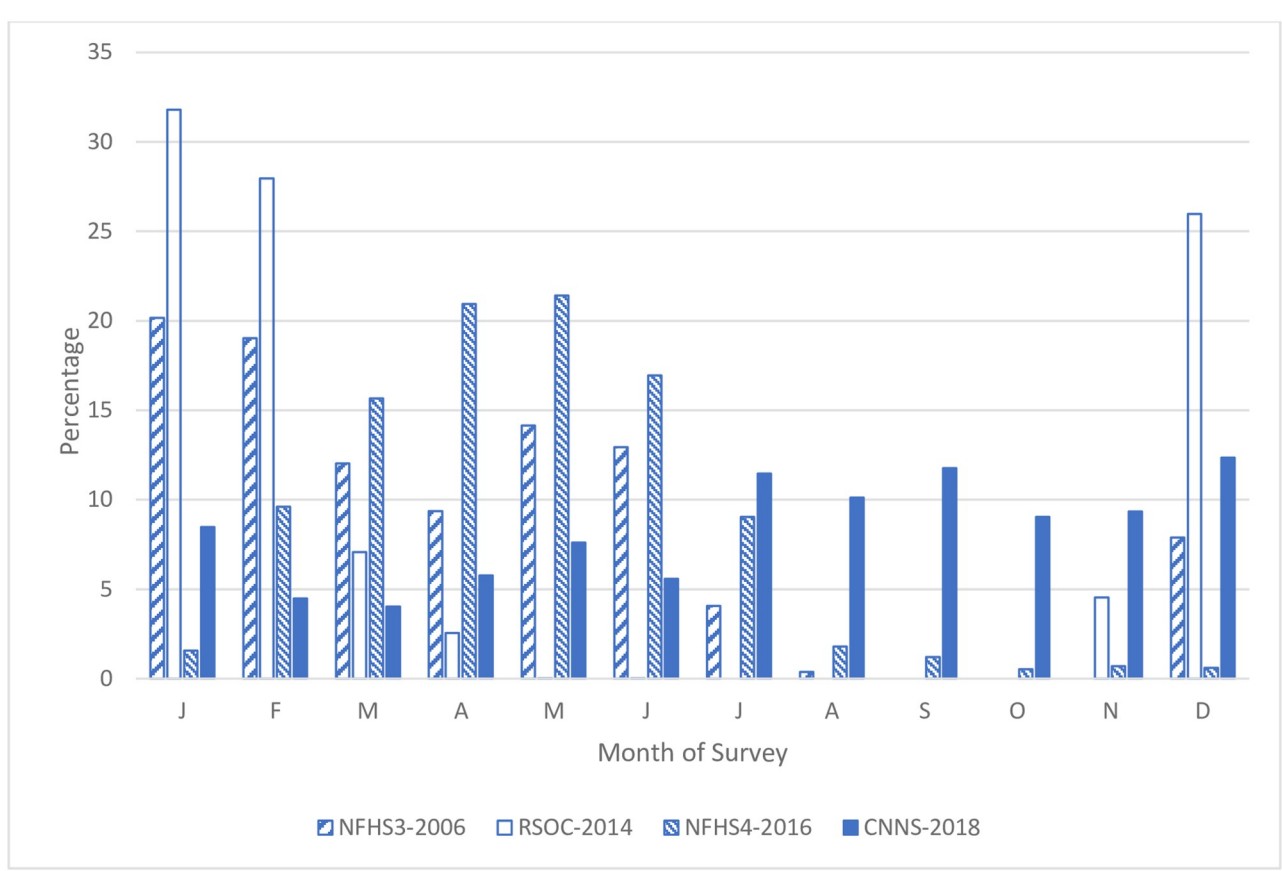

**Fig 1. Distribution of month of data collection by NFHS-3, RSOC, NFHS-4 and CNNS.**

January and February. The NFHS3 collected data from December 2005 until August 2006. The NFHS4 collected data from January 2015 until December 2016. For the NFHS 3 and 4, the majority of data was from January to May (over two-thirds). The CNNS with data collection from April 2016 to September 2018 was the most uniform with data collection from all months of the year. The period from July to September is commonly considered the peak season for acute malnutrition. In the RSOC survey no data were collected from June to September as compared to 5% in NFHS-3, less than 10% in NFHS-4 and 32% in CNNS. To make a robust comparison of a seasonally affected variable such as wasting, it is necessary to control for the timing and socio-demographic variation in data collection across the different surveys.

In this analysis, we examine the effect of month of data collection across the four surveys on the fluctuation in wasting prevalence. We develop regression models to estimate the mean WHZ and wasting prevalence for every month of the year for an average year adjusting for socio-demographic differences in data collection across month of data collection and survey. Using the estimates by month, we estimate the survey prevalence without effects of seasonal patterns and we present the trend along with the seasonal variation of wasting prevalence. This is the first analysis to present robust national monthly estimates of wasting and the comparable wasting estimates as a trend across four Indian surveys.

## Methods

### Data source and study population

Data from four nationally representative surveys designed to collect data on anthropometry in children under five years of age included in global Joint Malnutrition Estimates database [31] were used in these analyses. The NFHS-3 (2005–06) collected anthropometry from 46,655 children under five. The RSOC (2013–14) collected anthropometry from 90,908 children under five. The NFHS-4 (2015–16) collected anthropometric measures from 259,627 children under five. The CNNS (2016–18) collected data from 38,060 children under five. The NFHS-4 was designed to provide district-level estimates, while NFHS-3, RSOC and CNNS were designed to provide state-level estimates. The detailed methodology for the NFHS, RSOC and CNNS surveys has been published elsewhere [32–35]. After excluding the flagged or missing observations for weight-for-height z scores based on WHO 2006 growth standards (WHZ; <-5 or >5) and for missing data on month of interview, our initial sample contains information on 384,130 children under five years made up of 41,306, 82,544, 225,002 and 35,278 children from the NFHS-3, RSOC, NFHS-4 and CNNS, respectively.

### Variables

Our primary outcomes were mean WHZ scores in children <5 years, which indicates a child's growth in reference to a well-nourished child population [36] and wasting defined as WHZ < -2 SD. Across all the surveys, WHO 2006 growth standards were used to calculate the WHZ scores.

Studies in South Asia have shown that the highest prevalence of wasting in children under five years of age is found in the age group of 0 to 5 months [37]. Gender of the child is associated with the physical growth of the children. Boys are more likely to be slightly more wasted than girls [37]. Children who have experienced intrauterine growth retardation have a high risk of being undernourished at birth as well as wasted in the early childhood [38]. Maternal characteristics such as education level, health seeking behaviors and working status play an important role in the prevention of diseases like diarrhoea which affects the growth of the children [39]. Past research has shown that household level characteristics such as religion, caste, wealth and family affects the child's nutrition status [40, 41]. Consequently, we adjusted for

**Table 1. Summary statistics of the covariates included in the analysis by survey.**

| | Proportion (in %) | | | | | | | | |
|---|---|---|---|---|---|---|---|---|---|
| Variables | NFHS3 (2005–06) | RSOC (2013–14) | NFHS4 (2015–16) | CNNS (2016–18) | Variables | NFHS3 (2005–06) | RSOC (2013–14) | NFHS4 (2015–16) | CNNS (2016–18) |
| Month of Survey | | | | | Mother's education (in years) | | | | |
| January | 20.2 | 31.8 | 1.6 | 8.5 | No Schooling | 49.3 | 32.8 | 29.8 | 31.6 |
| February | 19.0 | 28.0 | 9.6 | 4.5 | Less than 5 years | 7.3 | 4.9 | 6.0 | 5.4 |
| March | 12.0 | 7.1 | 15.7 | 4.0 | 5–9 | 26.8 | 29.8 | 33.0 | 32.0 |
| April | 9.4 | 2.5 | 20.9 | 5.8 | 10–11 | 7.8 | 13.2 | 12.0 | 11.3 |
| May | 14.1 | 0.0 | 21.4 | 7.6 | 12 or more years | 8.8 | 16.1 | 19.1 | 19.6 |
| June | 12.9 | 0.0 | 16.9 | 5.6 | Don't know/ Missing | 0.0 | 3.3 | 0.0 | 0.1 |
| July | 4.1 | 0.0 | 9.0 | 11.5 | Mother's working status | | | | |
| August | 0.4 | 0.0 | 1.8 | 10.1 | Not working | 62.6 | 77.2 | 13.4 | 75.2 |
| September | 0.0 | 0.0 | 1.2 | 11.8 | Working | 37.4 | 19.4 | 3.7 | 24.6 |
| October | 0.0 | 0.0 | 0.5 | 9.0 | Don't know/ Missing | 0.0 | 3.3 | 82.9 | 0.2 |
| November | 0.0 | 4.5 | 0.7 | 9.3 | Religion | | | | |
| December | 7.9 | 26.0 | 0.6 | 12.3 | Hindu | 78.6 | 78.2 | 78.7 | 79.3 |
| Missing | 0.0 | 0.1 | 0.0 | 0.0 | Muslim | 16.7 | 16.6 | 16.5 | 16.0 |
| Sex | | | | | Christian | 2.0 | 2.6 | 2.0 | 2.6 |
| Male | 52.3 | 51.3 | 51.9 | 51.5 | Sikh | 1.4 | 1.3 | 1.3 | 1.0 |
| Female | 47.7 | 48.7 | 48.1 | 48.5 | Buddhist | 0.7 | 0.6 | 0.8 | 0.6 |
| Child's age (in months) | | | | | Other | 0.8 | 0.6 | 0.7 | 0.5 |
| 0–5 | 7.0 | 9.5 | 8.9 | 8.8 | Missing | 0.0 | 0.0 | 0.0 | 0.1 |
| 6–11 | 7.0 | 9.4 | 10.1 | 10.1 | Caste | | | | |
| 12–17 | 7.1 | 10.9 | 10.1 | 9.6 | Schedule Caste | 20.5 | 20.3 | 21.7 | 23.1 |
| 18–23 | 6.2 | 8.9 | 9.8 | 9.4 | Schedule Tribe | 9.4 | 11.4 | 10.3 | 13.0 |
| 24–29 | 6.8 | 11.4 | 9.9 | 10.5 | OBC | 40.5 | 40.1 | 44.3 | 39.4 |
| 30–35 | 6.2 | 9.7 | 9.7 | 10.1 | Other | 26.4 | 27.3 | 19.5 | 24.5 |
| 36–41 | 6.4 | 11.1 | 10.5 | 10.7 | Don't know/ Missing | 3.1 | 1.0 | 4.3 | 0.0 |
| 42–47 | 5.9 | 10.5 | 10.0 | 10.0 | Wealth | | | | |
| 48–53 | 6.2 | 10.1 | 9.7 | 10.9 | Poorest | 25.0 | 20.5 | 25.1 | 20.3 |
| 54–59 | 5.4 | 8.6 | 8.9 | 10.0 | Poorer | 22.2 | 20.5 | 22.1 | 20.2 |
| Missing | 35.7 | 0.0 | 2.3 | 0.0 | Middle | 20.2 | 20.5 | 19.9 | 19.9 |
| Birth weight | | | | | Richer | 18.4 | 20.4 | 18.3 | 19.8 |
| Average or more | 27.4 | 31.2 | 65.1 | 57.3 | Richest | 14.2 | 18.2 | 14.6 | 19.9 |
| Low | 7.3 | 7.1 | 13.8 | 13.2 | Residence | | | | |
| Not weighed at birth | 60.6 | 0.0 | 18.5 | 18.2 | Urban | 24.7 | 30.1 | 27.8 | 23.4 |
| Don't Know | 4.6 | 0.0 | 2.6 | 3.9 | Rural | 75.3 | 69.9 | 72.2 | 76.6 |
| Missing | 0.1 | 61.7 | 0.0 | 7.4 | | | | | |
| Total | 100.0 | 100.0 | 100.0 | 100.0 | Total | 100.0 | 100.0 | 100.0 | 100.0 |

these individual, maternal, household and community level characteristics as they are known to be risk factors of wasting (summarized in Tables 1 and 2). Missing data and special responses such as "do not know" in various variables were coded as a separate category to minimize loss of information and potential selection bias.

**Table 2. Summary statistics of mean and SD of WHZ and household size by survey.**

| Variables | Average | | | |
|---|---|---|---|---|
| | NFHS3 (2005–06) | RSOC (2013–14) | NFHS4 (2015–16) | CNNS (2016–18) |
| Mean WHZ | -1.02 | -0.53 | -1.03 | -0.97 |
| SD WHZ | 1.29 | 1.54 | 1.36 | 1.24 |
| Household Size | 7.00 | 5.94 | 6.56 | 6.44 |
| Observations | 41,306 | 82,544 | 225,002 | 35,278 |

Individual level covariates included in the analysis were: sex (male and female); age (in six month groups, missing) and birth weight (low [<2.5 kg], average or more [2.5 kg +], not weighed at birth, don't know, missing).

The maternal level covariates included were: mother's education (no schooling, 1–4 years, 5–9 years, 10–11 years, 12 or more, don't know/missing) and mother's working status (in work force, not in work force, don't know/missing).

The household level covariates included were: religion (Hindu, Muslim, Christian, Sikh, Buddhist\Neo-Buddhist, other, missing) caste (scheduled caste [SC], scheduled tribe [ST], other backward class [OBC], and other), wealth status (in quintiles) and number of household members.

The community level covariate included was residence (urban or rural).

## Statistical analysis

We estimated the unadjusted prevalence of wasting by four surveys. These estimates were considered biased due to non-uniform distribution of the data collection over different months in the different surveys (Fig 1). In order to predict prevalence estimates across all months for all surveys, we pooled all datasets noting that the NFHS-4 and CNNS collected data from all calendar months.

Estimates of wasting prevalence are considered free of seasonal bias if the data are collected in the same season (or ideally in the same month). To account for bias in the prevalence estimates of wasting, we performed a series of unadjusted and adjusted Ordinary Least Squares (OLS) and logistic regression models using the pooled dataset of 384,130 observations from the four surveys. For this analysis, we included state fixed effects to control for time-invariant differences across the different states, which might be correlated with other explanatory variables. Sampling weights were used in the estimation of coefficients, mean WHZ, and prevalence of wasting. All analyses were performed in STATA, version 16.0 (Stata-Corp, College Station, TX).

To account the effect of seasonal patterns for overall estimates of wasting and WHZ by survey, we controlled for timing of data collection using month dummy variables along with the selected covariates. The results of the region-wise analysis were presented in S1 Appendix. To test the robustness of these estimates, we calculated the midpoint between peak and trough of monthly estimates from the logistic and OLS regression models controlling for background variation in data collection. For the RSOC data, where the peak and trough could not be calculated from the limited number of months of data collection, we used the pooled data to estimate the midpoint between peak and trough considering all data were RSOC over each month of the year. The adjusted estimates were approximately the same as the midpoint between peak and trough for all surveys.

To present the linear trends of wasting and WHZ from 2005 to 2018, we use the national weighted estimates with the effect of seasonal patterns accounted and plotted on the mid-point month of data collection for each survey. As the simple linear trend conceals the monthly

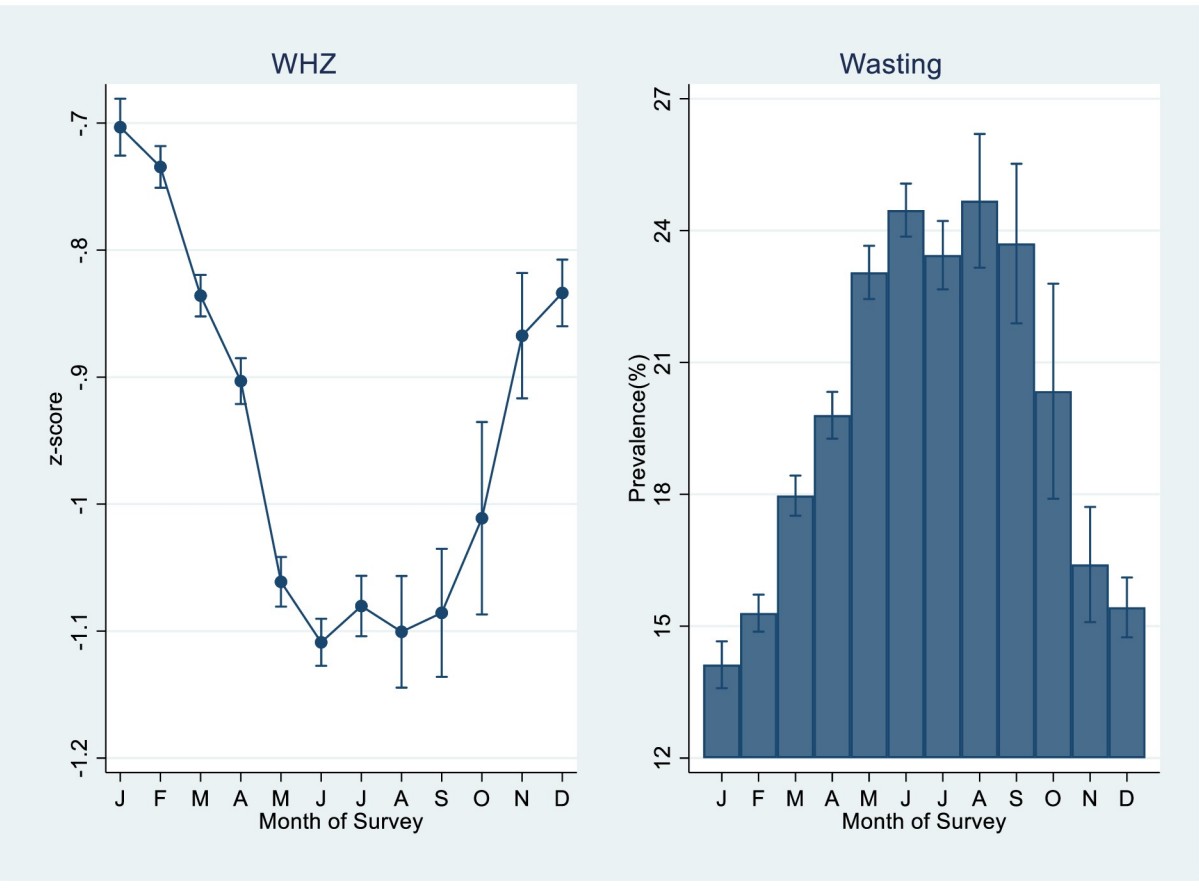

**Fig 2. Estimated mean WHZ and prevalence of wasting by month of survey from pooled data of four surveys.**

fluctuations in the data, we present a fitted seasonal trend in the wasting prevalence representing probable seasonal trends for comparison purposes. The fitted seasonal trend for wasting was calculated as follows:

$$A_{my} + [\cos\{(\text{month} - 7)(\pi/6)\}] * \text{amplitude of wasting} \tag{1}$$

where '$A_{my}$' is the linear estimate of wasting prevalence in the month 'm' of the survey year 'y'. Month-7 adjusts the phase for the start of the wave. We assume that the peak of wasting is in July and the trough in January. Amplitude (5.3) is measured by the average of the difference between the highest (24.7% in August) and lowest (14.1% in January) adjusted wasting prevalence estimated by the pooled data (see Fig 2). The fitted seasonal trend from Eq (1) shows the expected fluctuations in wasting prevalence from January 2005 to December 2018.

To compare the survey results to the fitted trend, we calculated the adjusted prevalence estimates by month of survey for each survey. For the two surveys with data collection longer than one year, we present 12 months of seasonal fluctuation not accounting for interactions between month and year of data collection. We plotted the adjusted prevalence estimates by the month of survey for each survey to compare the fitted seasonal trend with the adjusted monthly wasting prevalence.

We fitted a similar seasonal trend for WHZ using Eq (2). Setting the inverse timing for Z-score with the assumption of the peak in January and the trough in July.

$$A_{my} + [\sin\{(month - 10)(\pi/6)\}] * \text{Amplitude in WHZ} \qquad (2)$$

We estimated the mean WHZ by month of survey using adjusted OLS regression model including month of survey along with the individual, maternal, household and community level characteristics for every survey separately. We used the pooled dataset of four surveys to estimate the mean WHZ (wasting prevalence) using adjusted OLS (logistic) regression model including month of survey and categorical variable identifying the survey (1 if NFHS-3, 2 if RSOC, 3 if NFHS-4 and 4 if CNNS) along with the individual, maternal, household and community level characteristics.

## Results

In the pooled data, the mean WHZ score was at its peak in January and its trough in June at -1.10 (95% CI, -1.13 to -1.09) but remained in the lowest range of -1.09 (95% CI, -1.14 to -1.04) until September before starting its ascent (Fig 2). We estimated the wasting prevalence using the logistic regression model including all the covariates of previous analysis. It was highest in the month of August with 24.7% (95% CI, 23.2% to 26.2%) if all the observations were treated as if they were collected in the month of August during their respective surveys (Fig 2). It was lowest (14.1%; 95% CI, 13.6% to 14.7%) if all the observations were treated as if they were collected in the month of January. Predicted wasting prevalence increased if all the observations were assumed to be collected in the months after January. For instance, it increased to 24.5% (95% CI, 23.9% to 25.1%) if all the individuals were treated as if they were collected in June. It declined to (15.4%; 95% CI, 14.7% to 16.1%) if we considered that all the observations were collected in December.

Analysis based on the NFHS-3, NFHS-4 and CNNS data indicate a U-shaped trend in the estimated mean WHZ by the month of survey except for in the RSOC data (Fig 3). Predicted wasting based on the adjusted logit regression analysis including all the covariates show that highest level of wasting is observed in the summer months across all the surveys. The peak varies from 28.0% (95% CI, 24.2% to 31.9%) in July in NFHS-3, 26.2% (95% CI, 24.4% to 28.0%) in August in NFHS-4 to 23.0% (95% CI, 19.5% to 26.5%) in July in CNNS if all the observations are treated as if they were collected in that particular month and survey.

The amplitude of seasonal variation in wasting and WHZ is presented in Table 3. The results are based on the findings from the months of data collection of each survey. In the NFHS 3, the data from 9 months of the year were adequate to identify the peak and trough. In the RSOC data, an un-biased estimate of amplitude cannot be calculated as majority of data were collected from three winter months. The amplitude of wasting steadily declines from the NFHS-3 to the CNNS from 7.6 to 5.8, with a corresponding decline in SD of WHZ (Table 2). The amplitude of more robust measure of WHZ finds the same measures of amplitude in NFHS-3 and CNNS indicating no reduction over 13 years. The WHZ measure of amplitude of NFHS-4 of 0.28 could indicate that the seasonal effect on WHZ during 2015–16 was more severe.

Using month of survey along with individual, maternal, household and community level characteristics in the adjusted regression analysis, mean WHZ would be -0.65 (95% CI, -0.68 to -0.63) instead of -0.53 (95% CI, -0.54 to -0.51) if all observations in the data were treated as if their data was collected in RSOC (Fig 4). It would be -0.98 (95% CI, -1.00 to -0.97) and -0.93 (95% CI, -0.96 to -0.90) instead of -1.03 (95% CI, -1.03 to -1.02) and -0.97 (95% CI, -0.99 to -0.95) if all observations in the data were treated as if their data was collected in NFHS-4 and

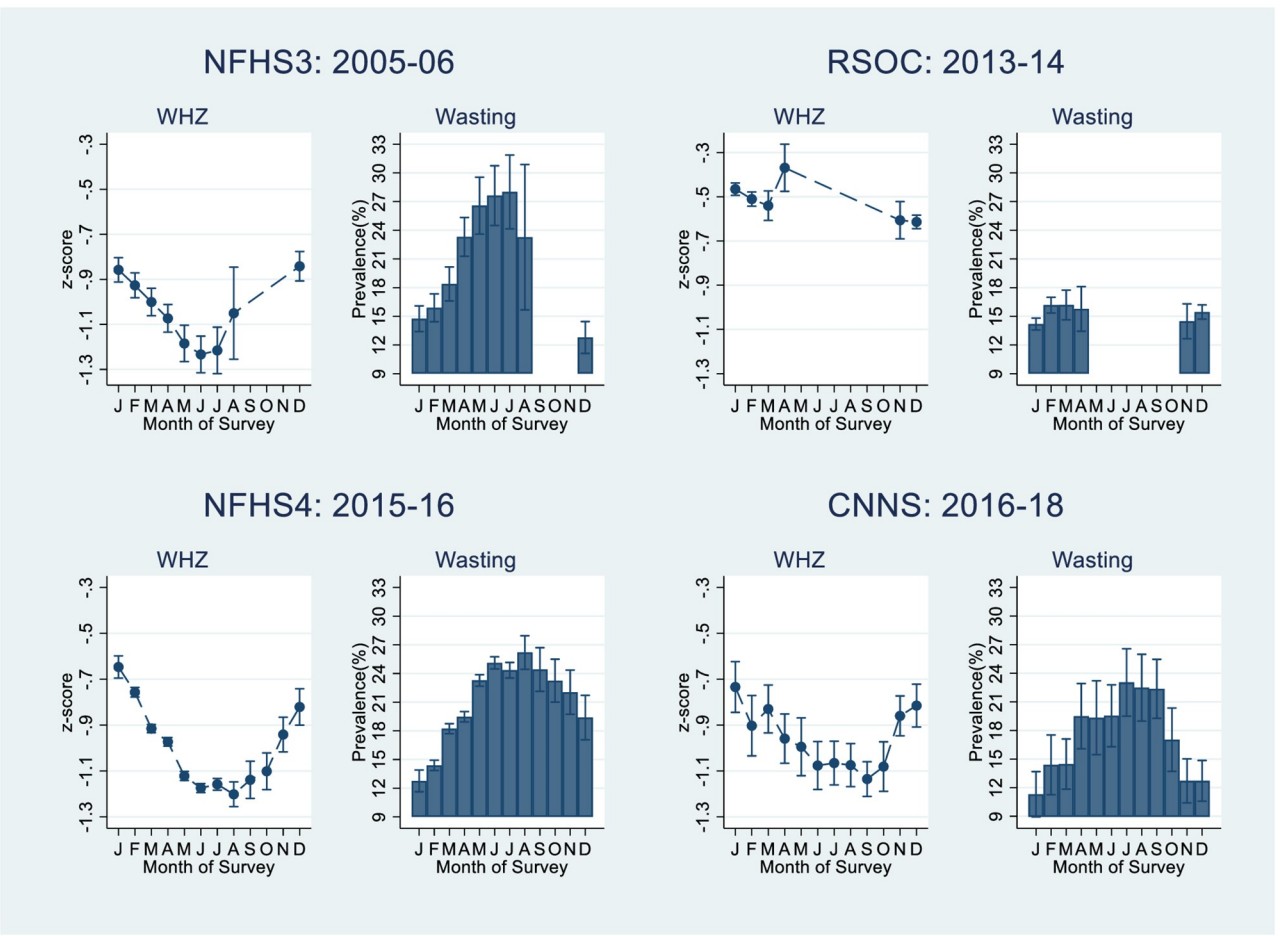

**Fig 3. Estimated adjusted mean WHZ and prevalence of wasting presented by month of survey from NFHS-3, RSOC, NFHS-4 and CNNS.**

CNNS, respectively. Correspondingly, we found that predicted wasting prevalence estimated through adjusted logistic regression analysis would be 18.6% (95% CI, 17.8% to 19.3%) instead of 15.2% (95% CI, 14.9% to 15.6%) if all observations in the data were treated as if their data was collected in RSOC (Fig 4). Similarly, it would be 19.7% (95% CI, 19.3% to 20.0%) instead of 21.0% (95% CI, 20.8% to 21.3%) if all individuals in the data were considered as if their data was collected in NFHS-4. No significant differences in predicted wasting prevalence if everyone in the data were treated as if their data was collected in NFHS-3 or CNNS.

The overall wasting prevalence unadjusted for seasonal patterns in the four surveys made a jagged pattern over the period of 2005–2018. The unadjusted wasting prevalence decreased from 19.8% in 2005–06 (NFHS-3) to 15.2% in 2013–14 (RSOC) followed by increase to 21.0%

**Table 3. Amplitude of seasonal variation of prevalence of wasting and WHZ by survey.**

| | Amplitude of seasonal variation | | | | |
|---|---|---|---|---|---|
| | **NFHS3** | **RSOC** | **NFHS4** | **CNNS** | **Estimated from pooled Data** |
| Wasting | 7.6 | 0.1 | 6.7 | 5.8 | 5.3 |
| WHZ | 0.20 | 0.12 | 0.28 | 0.20 | 0.20 |
| Observations | 41,306 | 82,848 | 225,002 | 35,278 | 384,130 |

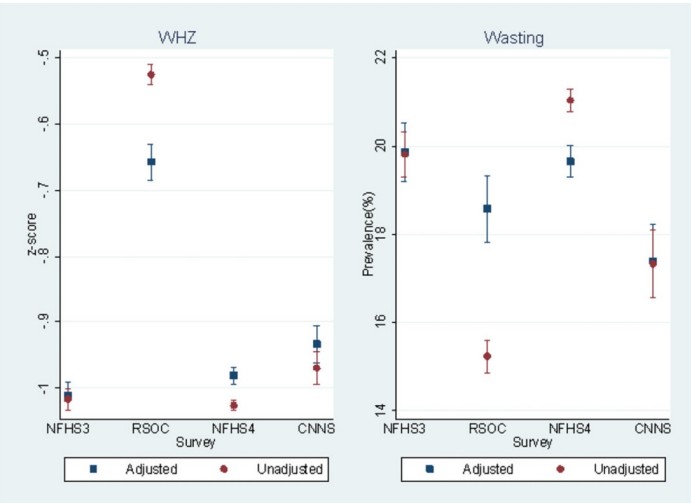

**Fig 4. Estimated mean WHZ and prevalence of wasting by month of survey from pooled data of four surveys with adjusted and unadjusted estimates.**

in 2015–16 (NFHS-4) and a decline to 17.3% in 2016–18 (CNNS) (Unadjusted estimates in Fig 4). These comparisons are considered biased due to differences in timing of data collection across different surveys.

Even with adjustments on WHZ for the RSOC, the data collected from 4 months of data collection in the winter season shows the strong seasonal effect. In the analysis of adjusted wasting prevalence over the NFHS 3, RSOC and NFHS 4 surveys, there was overlap in the confidence intervals indicating no difference between the adjusted estimates over the 11 years. The standard deviation of WHZ in the RSOC helps to interpret the differences in WHZ and no significant difference found in the prevalence.

In S2 Fig, we use the adjusted estimates of wasting prevalence accounting the effects of seasonal patterns (19.9% in 2005–06, 18.6% in 2013–14, 19.7% in 2015–16, and 17.2% in 2016–18) to track the linear trend of wasting from January 2005 to December 2018. With the amplitude from the overall pooled survey data, we fit a seasonal trend. From the individual surveys, the adjusted monthly estimates closely overlap the fitted trend for every survey round. Similar seasonal trend for WHZ is presented in S3 Fig using Eq (2).

The unadjusted trend of wasting prevalence between 2006 and 2018 showed a significant decrease from NFHS-3 to RSOC followed by increase with NHFS-4 and decline with the CNNS (S2 Fig). The jagged trend line is biased by the non-uniformity of data collection and not planning for the effects of seasonal patterns in the survey implementation (Fig 1). Our analysis based on the adjusted OLS regression model indicate a non-linear trend in the estimated mean WHZ by the month of survey. This is reflected in the estimated wasting prevalence by the month of survey (Fig 3).

Analysis based on the pooled dataset of four surveys NFHS-3, RSOC, NFHS-4 and CNNS suggests that mean WHZ is highest in January, falls to its trough in the summer months (June to September) and returns to its peak in the winter months (Fig 2). This analysis suggests that estimated mean WHZ decreased and wasting prevalence increased adjusting for the effect of month of survey along with individual, maternal, household and community level characteristics in NFHS-4 and CNNS (Fig 4). Fitted seasonal trend indicates that estimation of wasting prevalence and WHZ at the national level averages miss out the effect of seasonal patterns (S2

and S3 Figs). When seasonal patterns are not controlled for in data collection or analysis, estimates of wasting at state and country levels are likely to be biased/misrepresented, with potential implications for misguided policy and programming.

## Discussion

The effects of seasonality on wasting are well documented. Current large scale survey methods are not designed to collect data on seasonality. Using survey data with representation from all twelve months of the year in one or a group of surveys, it is possible to analyse seasonal patterns. With these improved analysis methods, trends with the effects of seasonal patterns accounted and robust monthly estimates of seasonal affected prevalence can be calculated from cross-sectional survey data.

Strengths of our analysis are estimates based on four cross-sectional nationally representative surveys demonstrating similar patterns of seasonal patterns in each round. The survey estimates removing the effect of seasonal patterns are calculated directly from individual surveys for three of the four surveys. For the remaining survey, the pooled dataset of four datasets used to estimate the estimates removing the effect of seasonal patterns using statistical methods normally used to calculate estimates from pooled datasets for domains with no acceptable valid data.

When analysing the seasonal trends of wasting, no assumptions were made about pattern of wasting prevalence or WHZ across months of the year. Following the adjustments for the socio-demographic differences in data collection by month, clear seasonal trends are apparent in the 2005 data and these continue up until the CNNS 2016–18 data. Also, with the estimates of wasting accounting the effects of seasonal patterns to make comparisons across surveys, no assumptions were made concerning which month(s) might be the most or least affected by seasonality.

The weakness of our study lies in the fact that pooled repeated cross-sectional surveys could not ensure that the timing of data collection from different states is uniform across the surveys. Cross sectional surveys repeated in the same months across the different states would be helpful in providing the progress made at the state and national level during that specific month. Moreover, dynamic nature of variation in wasting over time both at the individual as well as population level is not well captured in a cross sectional setting [23].

Our results demonstrate that seasonality must be controlled for in the planning, data collection and/or analysis. It is recommended that surveys must collect data during the same month every year and control for seasonality in the analysis and presentation of results. In west Africa, by design annual surveys were conducted in July to control for effects of seasonality for cross country and cross year of data collection comparison.

In comparisons with other data from India and the region, similar seasonal trends are found when assessing the effect of seasonality on wasting. The long running HKI surveillance system collected representative samples of child malnutrition by month in Bangladesh from 1990 to 2000. The seasonal variation of wasting ranged from 10% in December and January up to its peak in August at about 20% [42]. Similarly, in a study of tribal populations in Odisha, 16.5% of under five children were found to be significantly more wasted in the rainy season (June to September) compared to 10.6% in the dry season (February to May) [43].

In a South Asia regional analysis of determinants of wasting, seasonal patterns was not consistently associated with wasting in the final models [37]. This likely due to persistent high prevalence of wasting in all seasons (10% or more in all countries) along with non-specific and reductive methods to analyse seasonal patterns (one variable identifying dry or monsoon season).

Admissions for management of severe acute malnutrition in India are also seasonally affected. The Community-Based Management of Severe Acute Malnutrition program in Bihar, India (MSF- BIHAR) reported with data from 2009 to 2011 that peaks in admission fell in the months of June to August [44]. In Southern Rajasthan, a hospital based malnutrition treatment centre site from data in 2014–16 reported highest admissions from July to September [45]. Even in urban areas such as Delhi, evidence from 2012–2014 of seasonal patterns in admissions for SAM treatment was found with the greatest numbers of admissions with SAM from May to October [46].

A global review of longitudinal studies on wasting including ten cohorts from south Asia found the troughs of WHZ and peaks of wasting in all cohorts were found during the peak of rainy season. Seasonal variation ranged between 0.3 to 0.5 WHZ scores in south Asian cohorts [23]. This is similar to what is reported above with an overall 0.4 WHZ scores variation across the four surveys.

The seasonal nature of wasting demands application of methods during the design, implementation and/or analysis to control for temporal variation to ensure meaningful comparisons. These controls must be used with comparisons within (for validation of results) and across years (trends), across geographies and for determinant/decomposition analyses and burden calculations.

A majority of LMIC countries implement growth monitoring and promotion (GMP) programs to track both individual and aggregate growth with the objective to identify and address failure to thrive. These large scale efforts are often affected by error introduced by quality of tools, methods and reporting processes. The programs are commonly evaluated with more robust survey data. It is critical to generate data to support or validate these GMP programs, that the survey results must control for seasonality prior to any comparisons.

Comparisons across geographies are important to identify significant changes. A major step forward in tracking malnutrition has been the global mapping methods introduced by Simon Hays [47]. As these maps are based on cross-sectional data, adjustments for viable comparisons across countries and time are needed. In the mapping methodology, it was noted the seasonal patterns adjustment had relatively little effect on raw data. The wasting observations were adjusted to a mean day within a month of a 12 month periodic spline. If all estimates were presented with the effects of seasonal patterns removed, the seasonal effects on the raw data would likely be more pronounced.

In depth studies on the causes and explanation of change of wasting must also control for the effects of seasonal bias. Determinant and decomposition analyses that do not effectively control for seasonality in the analysis will likely produce anomalous results.

Survey measures of wasting must remove the effect of seasonal patterns before making robust analysis of trends. The 2019 WHO/UNICEF report providing recommendations for the data collection of anthropometric indicators in children under-five recommends survey planning to "identify the best period to implement the survey to allow comparison with previous surveys" but does not describe methods to control for seasonal bias [48].

Improved methods for trend analysis of global wasting are needed. Current tracking efforts present trends for stunting and overweight but only point prevalence and burden of wasting. The point estimates are calculated from national survey estimates collected over different months and years assuming that all effects of seasonality dissipate within the aggregate estimate.

Global trends of wasting need better methods to remove the effect of seasonal patterns to present national estimates for comparison across geographies and time. Global overweight trends are calculated and presented. As overweight is based on the same inputs as acute malnutrition (WHZ), the seasonally biased trends will also benefit from application of methods to control for seasonal patterns in planning, implementation and analysis.

As wasting is a temporal condition like many illnesses, cross-sectional surveys cannot measure timing of onset of acute malnutrition or length of condition, which is needed to calculate the severity of conditions and burden. The number of wasting episodes within a year is incompletely captured in cross-sectional, survey-based estimates of wasting. In longitudinal studies, analyses showed that at two years of age, one-third of children had experienced wasting, which was 5 times higher than the prevalence estimate (6%) [23]. Burden estimates will continue to be a probable minimum estimate until the seasonal effects on incidence are considered.

Populations must be protected from seasonal deprivations that provoke acute malnutrition. Greater efforts are needed to ensure that social safety nets are reinforced in the season leading up to the peaks of acute malnutrition and provide the full range of entitlements needed to protect populations from these deleterious effects. To have more timely and accurate measures of seasonality of wasting and the subsequent effects on the burden of wasting in children under-five, improved, regularly conducted, cost-effective methods are needed to effectively eliminate seasonal threats to children and to reduce incidence of wasting to its minimal acceptable level.

## Supporting information

**S1 Fig. Unadjusted estimates of prevalence of wasting by month of data collection in NFHS-3, RSOC, NFHS-4 and CNNS surveys.**
(TIF)

**S2 Fig. Seasonal trend and prevalence of wasting by NFHS-3, RSOC, NFHS-4 and CNNS.**
(TIF)

**S3 Fig. Seasonal trend and estimates of WHZ scores by NFHS-3, RSOC, NFHS-4 and CNNS.**
(TIF)

**S4 Fig. Estimated mean WHZ and prevalence of wasting by month of survey from pooled data of four surveys with adjusted and unadjusted estimates: Region wise analysis.**
(TIF)

**S1 Appendix.**
(DOCX)

**S1 File.**
(DTA)

**S2 File.**
(DTA)

## Author Contributions

**Conceptualization:** Robert Johnston, Mudit Kapoor, Praween K. Agrawal, Arjan de Wagt.

**Data curation:** Gaurav Dhamija.

**Formal analysis:** Gaurav Dhamija, Mudit Kapoor.

**Methodology:** Robert Johnston, Gaurav Dhamija, Mudit Kapoor.

**Supervision:** Robert Johnston, Mudit Kapoor, Praween K. Agrawal, Arjan de Wagt.

**Validation:** Robert Johnston, Gaurav Dhamija, Mudit Kapoor.

**Visualization:** Gaurav Dhamija.

Writing – **original draft:** Gaurav Dhamija.

Writing – **review & editing:** Robert Johnston, Mudit Kapoor, Praween K. Agrawal, Arjan de Wagt.

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
