## [Decision Letter · Decision Letter 0]

22 Jun 2021

PONE-D-21-13920

Methods for assessing seasonal and annual trends in wasting in Indian Surveys (NFHS-3, 4, RSOC & CNNS)

PLOS ONE

Dear Dr. DHAMIJA,

Thank you for submitting your manuscript to PLOS ONE. After careful consideration, we feel that it has merit but does not fully meet PLOS ONE’s publication criteria as it currently stands. Therefore, we invite you to submit a revised version of the manuscript that addresses the points raised during the review process.

I think you ask a very important question, and given the debates surrounding "wasting" in India, this kind of careful research is extremely useful. But as both the reviewers indicate, it will be important to delve deeper into the question of seasonality in the measurement, and the dates when the surveyors actually visited the households in different regions. There are also questions for how you measure "bias". Ultimately, you may want to answer whether the existing survey data allows for sensible comparison across surveys at all.  Please do also provide details on the access to the RSOC data, and how the data may be available to other researchers. The document also needs proof-reading.

We look forward to receiving your revised manuscript.

Kind regards,

Renuka Sane

Academic Editor

PLOS ONE

Journal Requirements:

 All authors received joint funding working at UNICEF and ISI.

There is no Grant Number

Aditya and Megha Mittal

Funders don't have a website.

No role played by the funders

We note that one or more of the authors is affiliated with the funding organization, indicating the funder may have had some role in the design, data collection, analysis or preparation of your manuscript for publication; in other words, the funder played an indirect role through the participation of the co-authors. If the funding organization did not play a role in the study design, data collection and analysis, decision to publish, or preparation of the manuscript and only provided financial support in the form of authors' salaries and/or research materials, please do the following:

a. Review your statements relating to the author contributions, and ensure you have specifically and accurately indicated the role(s) that these authors had in your study. These amendments should be made in the online form.

b. Confirm in your cover letter that you agree with the following statement, and we will change the online submission form on your behalf: 

“The funder provided support in the form of salaries for authors [insert relevant initials], but did not have any additional role in the study design, data collection and analysis, decision to publish, or preparation of the manuscript. The specific roles of these authors are articulated in the ‘author contributions’ section.

5. Thank you for providing the following Funding Statement:  

All authors received joint funding working at UNICEF and ISI.

There is no Grant Number

Aditya and Megha Mittal

Funders don't have a website.

No role played by the funders 

We note that one or more of the authors is affiliated with the funding organization, indicating the funder may have had some role in the design, data collection, analysis or preparation of your manuscript for publication; in other words, the funder played an indirect role through the participation of the co-authors. 

If the funding organization did not play a role in the study design, data collection and analysis, decision to publish, or preparation of the manuscript and only provided financial support in the form of authors' salaries and/or research materials, please review your statements relating to the author contributions, and ensure you have specifically and accurately indicated the role(s) that these authors had in your study in the Author Contributions section of the online submission form. Please make any necessary amendments directly within this section of the online submission form.  Please also update your Funding Statement to include the following statement: “The funder provided support in the form of salaries for authors [insert relevant initials], but did not have any additional role in the study design, data collection and analysis, decision to publish, or preparation of the manuscript. The specific roles of these authors are articulated in the ‘author contributions’ section.” 

If the funding organization did have an additional role, please state and explain that role within your Funding Statement. 

Please also provide an updated Competing Interests Statement declaring this commercial affiliation along with any other relevant declarations relating to employment, consultancy, patents, products in development, or marketed products, etc.  

Reviewers' comments:

Reviewer's Responses to Questions

**Comments to the Author**

1. Is the manuscript technically sound, and do the data support the conclusions?

Reviewer #1: Partly

Reviewer #2: No

2. Has the statistical analysis been performed appropriately and rigorously? 

Reviewer #1: Yes

Reviewer #2: No

3. Have the authors made all data underlying the findings in their manuscript fully available?

Reviewer #1: No

Reviewer #2: No

4. Is the manuscript presented in an intelligible fashion and written in standard English?

Reviewer #1: No

Reviewer #2: Yes

5. Review Comments to the Author

Reviewer #1: This paper analyzes trends in wasting in India in 2005-2018 using data from four different consecutive cross-sectional surveys. I think the paper makes a simple but often forgotten point, that is, that estimates of wasting (an indicator that can change quickly) in LDCs are likely affected by seasonality, and because different surveys usually measure height and weight of children in different months in different locations, comparing levels over time without taking this into account may lead to misleading estimates of trends.

I think this is a useful paper although there are some issues that should be addressed or at least acknowledged.

- As highlighted by the authors, a drawback of the paper is that, in principle, seasonality may be different in different location. In particular the agricultural season and the monsoon vary a good deal between different states in a country as huge as India. On the one hand I see that it is impossible to take this perfectly into account, given that there are several states where one does not have observations for all months of the year. However perhaps a reasonable compromise would be to divide India in, say, 3 or 4 macro-areas and perhaps by sector. As long as every month is represented in each macro-area I think the procedure described here could be repeated within each separate macro-area. Right now you have state FE but this (as you recognize) does not allow for state-specific seasonality but only for state-specific shifters.

- An important point you must acknowledge is that (if I am not mistaken) you are assuming that seasonality patterns do not change over time. This is important for you, because if such patterns changed over time you would be back to square one, without a way to fix the problem. I also think that such assumption is problematic if one is looking at long periods of time, given that seasonality of wasting should be expected to decline over time with economic development.

- You should be more precise early on about what you mean by "bias". While your argument is intuitively clear, it is not completely clear what the "right" measure should be. E.g. in several points in the paper you refer to the % of children who have been wasted at least one in a year, or the total number of "wasting" episodes in a year, but neither (as you recognize) can be estimated without detailed surveillance data, and if I am not mistaken this is NOT what your method tries to recover (or is it?). So please early on be specific about (a) what is it that you would like to estimate IDEALLY (e.g. the % children who have been wasted at least once in a year) (b) what is it that your method allows to recover under your assumptions, even if it is a second-best (e.g. wasting in the middle month of the survey).

- I think some of the surveys you mention report height and weight for children 6-59 months old and not 0-59 ("under five"). Please double check.

- Please provide more details on which growth reference charts you have used to construct the z-scores. Are these the WHO 2006?

- A point you do not discuss at all is the comparability of measurement across different surveys. This may sound unimportant if one thinks that age, height and weight are never measured with error, but unfortunately they are. I would like to see histograms of height and weight by survey to be more confident of data quality (I am not worried about NFHS which I know well). I have seen large-scale surveys (not the NFHS) were height and weight were measured very poorly, with clear peaks over certain focal values.

- In Table 1 you seem to indicate that age in months is missing for 36% of children in NFHS3, but I don't think this is correct. Please check.

- Top of P20 you write "Determinant and decomposition analyses that do not effectively control

for the bias in the analysis will likely suffer from attenuation bias," but while bias is likely it is far from clear that it will be "attenuation" bias given that seasonality can also lead to exaggerate certain patterns. Just to illustrate, suppose that socio-economic status predicts wasting only in certain months (e.g. the lean season): having data only from those months would exaggerate the "average" association one would find using data collected through the whole year.

- The manuscript would benefit considerably from a thorough professional proofreading as there are many instances (too numerous to list) where language is awkward and syntax incorrect.

Reviewer #2: This paper considers an important open question in the measurement of weight-for-age and wasting in India. It points out that child weight for height, and therefore the prevalence of wasting, depends on the season of measurement. It then asks: how should we compare weight-for-age and wasting measurements collected from surveys for which the season of measurement differs? This is especially relevant in light of the fact that although some other indicators of child health have shown improvement in successive health surveys, weight-for-height of children under five did not change between the NFHS-2005 and the NFHS-2015.

A main weakness of the paper is that it does not clearly define an “ideal” way to measure weight-for-height and then assess the possibilities for adjusting the available data to approximate the ideal. One proposal for an “ideal” weight-for-height measure might be the average weight-for-height of children under five in a random sample of children measured in every place in every month over a period of one year. Another possible “ideal” measure might be to estimate the highest month’s prevalence of wasting by visiting every place in that month. No matter what “ideal” gets picked, it would be useful to clearly define the object the adjusted average weight-for-height is trying to approximate.

It seems that not matter how the authors define an ideal way of measuring population weight-for-height, adjustments that can be made to the data will be limited by the fact that there is strong correlation between place and month of data collection (at least in the NFHS data). Although the paper makes brief mention of this correlation, and claims to deal with it by using state fixed effects in regression specifications, this approach may not be adequate. To give an extreme example, if all of the data for Bihar were collected in July, and all of the data from Kerala were collected in January, there would be no overlap between the months of data collection and the places of data collection, and so it would not be possible to create an adjusted estimate that averages over the entire year and the entire country. Nor would it be possible to compute a national estimate for July; nor for January. In order to make the case that their adjustments are sensible, the authors need to show that there is enough data collected from each state in each month (or from all states in a given month).

Perhaps a different approach for this paper would have been to ask: is it the case that we can create meaningful comparison of weight-for-age across surveys? The answer may be that it is not possible to do this for India as a whole. That said, it may be possible to compare across surveys for particular states if they happened to be surveyed in the same month by different surveys.

A revised paper might propose an “ideal” way of measuring weight-for-height, explain what the available data allow us to measure, and explains the adjustments the authors make to try to approximate the ideal. It would also be useful to mention how the adjustments fall short of the achieving the ideal measurement.

Here are some minor comments on presentation:

- The use of the term “random” has been used to refer to statistical randomness in some places in the text and to arbitrariness in others.

- It would be good to put the years on the survey labels in Table 1.

- Among the four surveys, only the RSOC has a very different unadjusted WFH than the others. It may be useful for the authors to put more focus on explaining why the RSOC is different from the other surveys.

- It may not make sense for Figure 4 to be “connected,” as it does not show a time trend.

- I would suggest not including Figures 5 and 6 as the authors have not done much to show evidence for the cyclical pattern that the figures assumes.

6. PLOS authors have the option to publish the peer review history of their article (what does this mean?). If published, this will include your full peer review and any attached files.

Reviewer #1: No

Reviewer #2: No

---

## [Author Response · Author response to Decision Letter 0]

19 Aug 2021

We have attached the response to academic editor report and response to reviewers' report.

---

## [Editor Report · Decision Letter 1]

8 Nov 2021

Methods for assessing seasonal and annual trends in wasting in Indian Surveys (NFHS-3, 4, RSOC & CNNS)

PONE-D-21-13920R1

Dear Dr. DHAMIJA,

We’re pleased to inform you that your manuscript has been judged scientifically suitable for publication and will be formally accepted for publication once it meets all outstanding technical requirements.

Kind regards,

Renuka Sane

Academic Editor

PLOS ONE
---

## [Editor Report · Acceptance letter]

12 Nov 2021

PONE-D-21-13920R1 

Methods for assessing seasonal and annual trends in wasting in Indian Surveys
(NFHS-3, 4, RSOC & CNNS) 

Dear Dr. Dhamija:

I'm pleased to inform you that your manuscript has been deemed suitable for publication in PLOS ONE. Congratulations! Your manuscript is now with our production department. 

Kind regards, 

on behalf of

Dr. Renuka Sane 

Academic Editor

PLOS ONE